# Predictive Correlation between Apparent Sensory Properties and the Formation of Heterocyclic Amines in Chicken Breast as a Function of Grilling Temperature and Time

**DOI:** 10.3390/foods9040412

**Published:** 2020-04-02

**Authors:** Dániel Pleva, Katalin Lányi, Lívia Darnay, Péter Laczay

**Affiliations:** University of Veterinary Medicine, Budapest 1078, Hungary; lanyi.katalin@univet.hu (K.L.); darnay.livia@univet.hu (L.D.); laczay.peter@univet.hu (P.L.)

**Keywords:** chicken meat, heterocyclic amines, harmane, norharmane, 3,8-dimethyl-imidazo(4,5-f)quinoxaline, 2-amino-3,4,8-trimethyl-imidazo(4,5-f)quinoxaline, 2-amino-1-methyl-6-phenylimidazo(4,5-b)pyridine, sensory analysis, colorimetric analysis

## Abstract

In the present set of experiments, we studied the correlation between the heterocyclic amine (HCA) concentration and the color changes of the chicken breast with or without skin during grilling under open or closed conditions as a function of the applied temperature and time. The concentration of the HCAs formed during grilling was measured by a validated LC–MS/MS method, whereas the color changes were determined either instrumentally or by visual observation. In general, higher temperatures and longer heat treatment times resulted in a more substantial HCA formation, especially on the surface of the samples and in the skin, where the total levels reached 746 ng/g. Results of regression analysis demonstrate a strong correlation (*r* > 0.7) between the HCA content of the grilled chicken breast and the L* and a* values indicating the significance of brightness and the red parameter of the color scale, respectively. In the case of open grilling, the skinless breast samples showed correlation (*r* > 0.7) between the HCA content and the color analysis results in both the full sample and the crust, respectively. Breast samples with skin exhibited the same level of correlation when they were grilled closed. In the case of open grilling the breast with skin, and closed-grilling the skinless breast, the linear regression analysis yielded a weaker correlation (0.7 > *r* > 0.4 or less) between the HCA concentrations and the color. Our results demonstrate that there is a predictive correlation between the color changes perceptible for the consumers and the HCA formation during grilling of chicken breast as a function of time and temperature depending on the type of grilling and the presence of skin.

## 1. Introduction

Food safety is a global issue that affects everyone. Several hazards complicate the work of the food chain safety authorities worldwide, because, although science evolves new technologies to keep these hazards under control, globalization and the expanding world population raise new problems. In addition, there is a part of food chain safety where the authorities have no strong influence, namely home cooking. Consumers often prepare their food at home from raw or half-ready ingredients, and that process is out of the scope of legal regulations. One of these home-made hazards is represented by the carcinogenic agents that are produced during heat treatment from different kinds of materials [1]. In our topic, chicken meat is in focus and the heterocyclic amines (HCAs) that are a group of potential carcinogenic side-products typically in meat dishes [2].

The HCAs that are important in food hygiene consist of two main groups depending on their structure and production circumstances [3]. The first group includes the thermic HCAs (aminoimidazoaroarenes), which are formed from amino acids, reducing sugars and creatine/creatinine generally over 150 °C as a side-product of the Maillard reaction. One member of this group, the 2-amino-3-methylimidazo(4,5-f)quinolone (IQ), was put on the 2A list (probably carcinogenic to humans) of the World Health Organization’s International Agency on Research of Cancer (WHO IARC) and several members are on the 2B (possibly carcinogenic to humans) list [4]. According to bibliographic sources, 3,8-dimethyl-imidazo(4,5-f)quinoxaline (MeIQx), 2-amino-3,4,8-trimethyl-imidazo (4,5-f)quinoxaline (4,8-DiMeIQx), and 2-amino-1-methyl-6-phenylimidazo(4,5-b)pyridine (PhIP) are the most frequent representants of this group in the chicken meat. That is why we chose to detect these three compounds for the examination of thermic HCAs in our experiments. The second group is the pyrolytic HCAs (aminocarbolines), which are produced during the pyrolysis of proteins and peptides on high temperature (generally over 300 °C). These compounds are not meat-specific; they can be found in protein-rich, heat-treated dishes of plant origin as well [5]. They are not considered to be directly carcinogenic, but they can increase the carcinogenic effect of major causatives, i.e., these compounds are co-carcinogens [6]. The two most common representatives of this group are 1-methyl-9H-pyrido(3,4-b)indole (harmane) and 9H-β-carboline (norharmane), so we also selected these compounds to characterize the formation of thermic HCAs [7].

Heat treatment is connected to color changes in the case of most foodstuffs. The most frequent reasons for these changes include the Maillard reaction (non-enzymatic browning) [8] and the pyrolysis of different compounds (burning)—i.e., the reactions that are responsible for the production of HCAs. Accordingly, the idea presents itself, whether there is a connection between the color formation of the meat and its HCA content. There are already bibliographic results available on the relation of instrumental colorimetric analysis with lipid oxidation in pork [9] or even with the HCA formation in smoked sausages [10], but no connection was tested with human senses. Buła et al. [11] designed an experiment for pork where HCA detection was combined with colorimetry and sensory analysis as well, but this sensory part did not contain color analysis. On the other hand, there are examples for HCA of combined color sensory tests that, however, lack instrumental verification [12]. Tikkanen et al. [13] performed a similar experiment with chicken meat, but they concentrated more on the mutagenic activity. Aasling et al. [1] also worked with chicken meat tested for HCA and color changes, but in this case, a specialist was employed for the color detection. Gibis and Weiss [14] tested the HCA formation and colorimetric changes in beef, but they only applied the L* (brightness) parameter from the instrumental colorimetry. Gibis and Loeffler [15] also used electric contact grill to roast chicken breast samples and instrumental colorimeter with human sensory supplement, but their aim was to observe the correlation between HCA formation and the initial sugar content of the meat samples.

The HCA content of chicken is not only dependent on the temperature of the heat treatment but also the duration has a major effect [16]. Besides those, the method of heat treatment [17] and the presence of skin on the surface of the meat [18] can modify the HCA production and also may have an effect on the degree of the color change.

In order to study the correlation between the HCA formation and the color changes of the meat, we performed a set of experiments applying an electric grill with two grilling plates that could be used in opened and closed state (model for different methods) and roasting the same kind of chicken breast with and without skin at different temperature and time combinations.

## 2. Materials and Methods

### 2.1. Chemicals

The HCA standards of MeIQx, 4,8-DiMeIQx, and PhIP were from Toronto Research Chemicals, harmane and norharmane were from Sigma-Aldrich, and caffeine was used as internal standard. Chemicals used for the sample preparation and the HPLC (acetonitrile, dimethylformamide, methanol, formic acid, acetic acid, and sodium hydroxide) were all purchased from VWR International. Type I ultra-pure water was produced by SUEZ Environment^®^ Water Purification System at our university.

### 2.2. Instruments and Equipment

For the grilling procedure, a DeLonghi CGH 1012D electric contact grill was used. The sample preparation for the chemical analysis was performed with the Department’s Bosch hand blender, Certomat WR water tub shaker, Biofuge Primo R centrifuge, and Biotage VP evaporator; the additional equipment was Phenomenex SI silicagel and Phenomenex C18 columns. For the analysis itself, a Shimadzu LCMS 8030 HPLC–MS/MS system was operated with a Phenomenex Kinetex C18 EVO 100 × 4.6 mm ID (2.6 µm particle size) column equipped with a 40 × 2 mm C18 guard column by LabSolutions^®^ software. The colorimetric analysis was carried out by a Konica Minolta CR-400 Chroma Meter equipment. Pure water was produced by the SUEZ Environment^®^ Water Purification System of the Department of Animal Hygiene.

### 2.3. Grilling Trials

For the experiments, Ross 308 chicken breast (filet and full breast) was purchased from a local retail market. The chemical analysis of this breed was done previously for amino acid profile and sugar content by the national accredited laboratory. For the grilling, standard size breast slices were prepared (40 grams of weight and 1.6 cm of thickness) with or without skin cover. Three slices were produced from every batch for the different analyses (sensory, colorimetry, HPLC). The grilling trials were performed at the Laboratory of Food Technology at our Department. The DeLonghi contact grill was utilized in opened and closed mode (in case of opened grilling, the meat slices were turned once, so the added time parameters are concerned with one side). Apart from the presence of the skin and the position of the top plate, the time and temperature of the heat treatment were also taken into account as parameters. The examined temperatures were 150, 190, and 230 °C, and the treatments lasted for 5, 10, or 15 min. The selection of these combinations was due to the expected lowest temperature of HCA formation (150 °C) [2], the upper limit of the temperature of the grill (230 °C), and our previous survey’s results [19,20]. In the case of closed grilling, the top plate was not flat but ribbed; the color measurement always happened from the meat part roasted on the flat side.

### 2.4. Sample Preparation for HPLC

From the grilled slices, 10 grams were taken as samples (both “inside” and “surface/skin”). The “inside” samples contained the inner, nonsuperficial part of the slices, and the “surface” (or in the case of skinned breast slices, “skin”) contained the superficial crusty layer only, where the color changes and the production of HCAs are more predictable. The samples were cut manually and homogenized, and then 2 grams from each were processed for the sample preparation. It started with shredding and homogenization by a hand blender, saponification with 15 mL of 1 M NaOH–water solution (skin samples with 25 mL of 5 M NaOH) and water tub shaking on 60 °C 190 rpm for 90 min (skin samples on 80 °C for 120 min). Then, it was taken into centrifuge tubes that were centrifuged 8000 rpm for 10 minutes on 10 °C (skin samples on 6 °C). The supernatant, enriched with 12 µL 50,000 ng/mL concentrated internal standard caffeine was taken onto Phenomenex Strata^®^ SI-1 Silica (55 µm, 70 Å), 500 mg/6 mL silica gel solid phase extraction (SPE) columns that were earlier conditioned by 2 mL H_2_O and 2 mL 1M NaOH solution. The sample was eluted by 2 × 2 mL ethyl acetate and then evaporated to dryness under N_2_ gas 50 °C in a Biotage VP evaporator. The same supernatant that had previously run down the silica gel SPE column was taken onto a Phenomenex C18 column that was pre-conditioned by 2 mL acetonitrile and 2 mL 1 M NaOH–water solution. The sample was eluted from the C18 by 2 × 2 mL acetonitrile into the same tube with the same numbered sample previously eluted and evaporated on silica gel; then, it was evaporated again the same way. There was some sludge remaining from the silica gel SPE that was resolved in 5 mL hexane, centrifuged on 8000 rpm 10 °C for 10 min. The supernatant was taken on silica gel SPE again, and the solution running out also got into the evaporator. The dry sample was dissolved in 0.6 mL acetonitrile.

### 2.5. Sensory Analysis

The sensory analysis was carried out by the coworkers at the university. They got a short training about this specific sensory analysis but no former education, so they represented the common consumers. For one examination, ten persons (6 females, 4 males; age 24–62 years) were always involved who had to test 3 samples at once in a laboratory room with standard 23 °C temperature, always at 11:00–11:30, before lunch. The samples were always covered by a random three-digit code, so the testers did not know what they studied. A sample took part of one standard slice in one piece (for the color analysis) and one slice of meat cut into cubes for the testers for the taste and texture examination. The testers had to add a value from 1 to 5 in the following parameters: Lightness–Brightness; Odor; Off-odor; Taste; Off-taste; Juiciness; Crunchiness; Overall impressions. The parameters and the scales were designed by a Ph.D. food engineer. For this article, only the Lightness–Brightness combined 1-to-9-value parameter is connected, where the values’ definitions are as follows:1—Black, burnt2—Mostly dark brown with burnt black spots3—Uniformly dark brown with no black spots4—Uniformly brown with no darker shades5—Golden brown (desired color)6—Uniformly golden yellow7—Uniformly light yellow8—Predominantly white with a slight yellow in some parts9—Whitish and raw

In the case of skinned meat samples, the skin-covered surface was measured. The answers were then analyzed, summarized and compared to each other.

### 2.6. Colorimetric Analysis

For the objective color analysis, a Konica Minolta CHROMA METER CHR-400 tristimulus color measuring system was used. The equipment measured the color of the surface in CIELAB values:L*—brightness, where value 100 means a perfect white and 0 is a total blacka*—redness, where the more positive the value, the redder the sample is (negative—green)b*—yellowness, where the more positive the value, the more yellow the sample is (negative—blue)

Ten individual measurements happened in each combination of parameters with the same external light conditions every time. In the case of skinned meat samples, the skin-covered surface was measured.

### 2.7. LC–MS/MS Analysis

Chemical quantitative analysis of HCA was performed by a Shimadzu LCMS 8030 HPLC–MS/MS system. Chromatographic separation was carried out on a Phenomenex Kinetex C18 EVO 100 × 4.6 mm ID (2.6 µm particle size) column equipped with a 40 × 2 mm C18 guard column. The gradient was designed with eluent A: 50 mM ammonium acetate in water (pH 5 adjusted with acetic acid) and B in 0.1 v/v% formic acid in acetonitrile. The flow rate was 0.4 mL/min, and one chromatographic run lasted 6 minutes. The column oven was set at 30 °C, and the autoinjector temperature was set at 7 °C. The injected volume was 10 µl. The quadrupole tandem mass spectrometer was used with electrospray ionization (ESI) ion source in positive mode and multiple reaction monitoring (MRM). Other MS parameters were: interface 4.5 kV; interface temperature 250 °C; desolvation line 300 °C; heat-block 350 °C; detector 1.78 kV; nebulizing gas (N_2_) 3 liters/min, drying gas (N_2_) 15 liters/min; collision gas (Ar) 230 kPa.

### 2.8. Validation of the LC–MS/MS Method

As a part of validation of the LC–MS/MS method, specificity/selectivity, linearity, limit of detection (LOD), limit of quantitation (LOQ), within- and between-run accuracy (precision and trueness) were determined for all five compounds tested. Specificity and linearity parameters were reliably met in all cases. The limit of quantitation was found to 0.25 ng/g for all compounds tested. The within-run and between-run precision values were well below the limit of 15%. The trueness as an indicator of accuracy of the method was also found to reliably meet the accepted limits between –20% and +10%.

### 2.9. Data Processing and Statistical Analysis

Data processing was carried out first by the LabSolutions^®^ software of the LC–MS system. Secondary data processing and part of the statistical analysis were made by MS Excel software (Microsoft, Redmond, WA, USA), for the correlation analysis using Pearson’s method.

## 3. Results

### 3.1. HCA Formation

During the quantitative chemical analysis, two pyrolytic (harmane, norharmane) and three thermic (MeIQx, 4,8-DiMeIQx, PhIP) HCAs were detected. The data set contains the individual results of the compounds and the summarized pyrolytic, thermic and total HCA amounts (Figure 1, Figure 2, Figure 3 and Figure 4). For each group, five heat treatment parameters were defined and tested: (1) presence of skin (with or without); (2) position of the top plate (opened or closed); (3) dividing the surficial parts from the inside of the slices (inside or surface/skin); 4 temperature (150, 190, or 230 °C); and (5) duration (5, 10, or 15 min).

The compounds and amounts of HCAs formed exhibited some differences based on the temperatures applied. While on 150 °C, the thermic ones, especially harmane, were typically the most frequently detected HCA component. On higher temperatures, PhIP dominated the HCA profiles: on 230 °C 15 min, it always made up of more than 50% of the total HCA ratio in the surface and the skin. MeIQx was usually the least detectable: in three cases of inside samples (opened without skin; closed with and without skin), its amount was even less than the limit of quantification (0.08 ng/g) at 150 and 190 °C. The formation did not seem to be significantly affected by the presence of skin or the top plate.

In the case of closed contact grilling, the HCA amounts were always higher compared to the opened samples grilled the same way. At 150 °C and 190 °C for 5 and 10 min, the inside samples contained less HCAs; if they were covered by skin, at higher temperature and/or longer time, this tendency changed. The skin always contained more total HCA than the same treated plain meat surface (Table 1, Table 2, Table 3, Table 4 and Table 5).

### 3.2. Color Changes by Grilling

#### 3.2.1. Objective Color Analysis

Out of the parameters of colorimetric measurement (L*, brightness; a*, redness; b*, yellowness), L* and a* showed a stronger correlation to the heat treatment time and temperature. Computing by time parameters, the trend lines’ R^2^ values exceeded 0.8 in 75% of the samples in the case of L* and also a*. In 92% of cases, there was a negative correlation for L* and a positive one for a*; however, the R^2^ was higher than 0.8 in 58% in the case of b* as well, but the correlation was 50% negative and 50% positive (Figure 5). The *p*-value was less than 0.1 in 42% of the cases of L* and 50% of the cases of a*, so we can admit a tendency in these cases.

#### 3.2.2. Subjective Color Analysis

For the sensory analysis, the value 5 was mentioned as the desired color of grilledness (golden brown). As expected, both the temperature and the duration of the heat treatment had an impact on the color of the grilled meat. The results obtained from the sensory checks confirm that the opened grilling method was actually milder, and their scoring values were usually lower than that given to the same level but to closed ones. The presence of skin had a braking effect on burning: the scoring values with skin were usually lower compared to the same results without skin, although the lowest value (2.8) was detected in the case of skinned breast, grilled closed, at 230 °C for 15 min (Table 6). In Scheme 1, a sample can be seen to compare the different colors of the samples without skin, grilled closed, on 190 °C for 5, 10, and 15 min.

### 3.3. Correlation Analysis

In the present experiments, three different data sets were compared: the chemical HCA analysis, the sensory color analysis, and the instrumental colorimetric analysis of meat samples heat-treated in the same way. By comparing these data sets, different edifications can be drawn. For the strength of correlation, the Pearson correlation coefficient (*r*) was determined. The correlation is considered strong if its absolute value (|*r*|) is higher than 0.7, and weak but demonstrable if it is between 0.3 and 0.7.

First, the results of human and objective color analysis were examined to determine if we could find any colorimetric parameters that are in close correlation with the sensory analysis. It turned out that the L* (brightness) and a* (redness) index had a very strong (L* *r* > 0.85; a* *r* < −0.9; *p* < 0.05) correlation in the cases of both the closed- and the opened-without-skin combinations. The opened-with-skin samples did not exhibit this correlation, which can be explained by the uneven color changes of the skin if the heat treatment is uneven and milder, for example, in the case of opened grilling. The b* (yellowness) index had a weak correlation with the sensory observations during opened grilling (−0.7 < *r* < −0.6; *p* < 0.05), and no connection was seen in the case of closed grilling. According to these results, L* and a* seem to be more related to the color changes perceived by the human eye, and the average |*r*| in their case is higher than 0.7, while for b*, it is just 0.34.

In addition, it was checked if there is any correlation between the objective instrumental colorimetric results and the HCA content of the meat determined. All five HCAs were individually examined, and the sum of the thermic, pyrolytic, and total HCA were considered. The correlation was only taken into account if there were at least five HCA results exceeding the limit of quantification. As regards the colorimetric parameters, L* and a* seemed to be the more useful ones. Their average |*r*| was higher than 0.3 for every single HCA, and for the totals, L* had an average *r* = −0.60 and a* *r* = −0.59 (*p* < 0.1). They were higher than 0.7 (*p* < 0.05) for the total HCA content at the closed-with-skin and opened-without-skin grillings, both with the inside and surface/skin results. For the chemical HCA analysis, in the case of closed-without-skin methods, only a weak correlation could be detected, and there was no correlation with the opened-with-skin combination. A correlation between the HCA content and b* could only be revealed in some cases, and it was never a strong correlation.

The third comparison was made was between the sensory analysis and the HCA content. In most cases, a strong correlation between subjective human observations and the amount of HCAs measured instrumentally was established. The average |*r*| value was higher than 0.7 (*p* < 0.05) in the case of every individual HCAs and for the thermic, pyrolytic, and total ones. Sorted by the grilling method, the worst R value was −0.64 (still *p* < 0.1) for the total HCA (closed-without-skin-surface), the weakest individual correlation (PhIP: opened-without-skin-inside) was −0.62. 

## 4. Discussion

Our HCA results are in harmony with previous bibliographic data. As was expected, PhIP was the most abundant HCA at higher temperatures, because its presence is the most common in the case of chicken meat [17,18,21]. It is a 2B category possible carcinogen on the IARC list; as a consequence, its high concentration may pose a hazard to consumers. The pyrolytic harmane and norharmane are not muscle-specific compounds, because their formation does not need creatine or creatinine. Accordingly, they can appear in other kinds of food as well [22]. Some former literature mentions that at least 300 °C is necessary for their formation [2], but recent bibliographic data [23,24] and our results also point out that these compounds can already be detected at temperatures as low as 150 °C.

To contextualize the toxicological significance of the measured HCA concentrations, we compared the exposure based on the poultry meat consumption data and the detected total HCA concentrations with the lower 95% confidence limit of the benchmark dose for 10% incidence of the carcinogenic effect observed in animals. For calculation of the exposure, we considered the mean per capita poultry meat consumption of 50.1 kg, i.e., 137 g/day reported by the Organisation for Economic Co-operation and Development for the USA as the highest level among the countries [25]. For PhIP, as the most abundant HCA detected at high temperatures, the lower confidence limit of the benchmark dose at 10% occurrence (BMDL10) was estimated at 2.72 mg/kg/day for colon tumors in rats and 0.48 mg/kg/day for prostate tumors [26]. Assuming a 70 kg body weight of a consumer, these values correspond to 189,700 mg/day and 33,600 mg/day of PhIP. 

As can be seen in Table 2, Table 3, Table 4 and Table 5, the highest concentration of PhIP measured in our experiments was 449.60 ng/g and 19.46 ng/g in the surface and the inside, respectively, of the chicken breast. Assuming a ratio of 1:9 between the surface and the inside parts, this would mean an average concentration of 62.74 ng/g. As a worst-case scenario, if we assume that the whole chicken meat is consumed as a grilled breast at 230 °C for 15 min, the estimated daily exposure to PhIP would be 8.60 mg/day. Based on the benchmark dose and the exposure presented above, the Margin of Exposure (MOE, the ratio of the benchmark exposure to the observed one) is 220,583,907 for colon and prostate tumors. In general, a margin of exposure of 10,000 or higher—if it is based on the BMDL10 from an animal study, and taking into account overall uncertainties in the interpretation—would be of low concern from a public health point of view [27]. Accordingly, the calculated MOE with regard to the colon tumor can be considered of low concern; however, the MOE for prostrate cancer may indicate a concern. In addition, the HCAs as tumor initiators can substantially increase the sensitivity of the organism to the simultaneously present tumor promoters that facilitate tumorigenesis. Therefore, their quantities should be reduced to the lowest possible levels [28].

Color analysis is an important part of food production and food quality. In addition, consumers’ choice can highly depend on the actual color of foodstuffs. Recently, more researchers highlighted its importance and relevance in relation to food safety as well, mainly by its connection with oxidation [9] or heat treatment [29]. 

Color measurement can be executed by instrumental or sensory tests. The instrumental analysis mostly operates by detection of the L*, a*, and b* (brightness, redness, and yellowness) parameters, and these results are objective [11]. Sensory analysis reports the consumers’ impressions better, and these tests can refer to the concrete color of the sample [12] or only its acceptance (hedonic sensory test) [14]. We found the first way better coupled with an instrumental analysis because, in applying this approach, these data were comparable.

In the context of hypothesizing a connection between the HCA content and the results of the color analysis, the sensory test exhibited the highest scores and strongest correlations with the measured HCA contents, followed by the L* and the a* values, respectively. Buła et al. [11] found a strong exponential relationship between 7,8-MeIQx and the b* value. This observation differs from our results, where b* correlation was found to be really weak, although we have not tested the same chemical compound. Our results exhibited a more pronounced correlation between the HCA content and the L* values compared to data reported by Gibis and Weiss for beef pate [14]. In the studies of Aaslyng et al. [11], the surface color was assessed on a five-point visual scale by using photographs of the meat samples. In order to reveal whether there is any connection between the HCA content and the surface color, percentage distribution of the content of HCAs related to the color was made. However, no correlation between the measured HCA content and the color changes was calculated. It was concluded that the HCA content was related to the surface color of the meat, and, especially in chicken, almost all samples with a dark surface color contained more HCAs than the lighter parts. This observation is in good agreement with our studies, where even the correlation between the measured HCA concentrations and the surface color can be demonstrated as a function of the applied temperature and time of grilling. Based on the results obtained from the present studies, the surface color of the chicken breast can be a useful indicator of the HCA content formed during grilling at different temperature and time.

## 5. Conclusions

The temperature and duration of the heat treatment had a major effect on the formation of HCAs, and the presence or absence of the skin also had a measurable effect. Closed contact grilling seems to be more effective, and the heat loss seemed to be lower, so the HCA levels were higher than in the case of opened grilling. The method of heat treatment also affected the color changes of the meat surface, both at the instrumental and the personal analysis. Comparing the data sets of these two fields, it is not startling that the results obtained from the present studies demonstrate a predictive correlation between the color changes perceptible for the consumers and the HCA formation during grilling of chicken breast as a function of time and temperature, depending on the type of grilling and the presence of skin. Results of regression analysis demonstrate a strong correlation (|*r*| >0.7) between the HCA content of the grilled chicken breast and the L* and a* values, indicating the brightness and the red parameter of the color scale. In the case of open grilling, the skinless breast samples showed correlation (|*r*| >0.7) between the HCA content and the color analysis results, both in the full sample and the crust. Breast samples with skin exhibited the same level of correlation when they were grilled closed. In the case of open grilling, the breast with skin, and in closed grilling, the skinless breast, the linear regression analysis yielded a weaker correlation (0.7 > |*r*| > 0.4 or less) between the HCA concentrations and the color.

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
