# Peer review of "Predictive Correlation between Apparent Sensory Properties and the Formation of Heterocyclic Amines in Chicken Breast as a Function of Grilling Temperature and Time"

_foods, 2020, doi:10.3390/foods9040412_

Round 1

Reviewer 1 Report

The article its interesting and presents some interesting results. The experimental design is simple and but objective and assertive, studing the effect of temperature and time, open and closure.

The abstract needs be improved including the p-value (significance) of the correlations indicated and could be interesting present more results about HCA, by effect of temperature and time (open and closed) (with values, not only text), the abstract must summarize the more important results of the work.

I suggest change the order of the keywords to: chicken meat, heterocyclic amines, colorimetric, sensory analysis,….

By the importance of sensory analysis that is an important objective to compare with HCA, the point 2.5 (line 137) must be improved and better explained. Which is the experience of the panel, ages, room conditions, hour of the taste, the methodoly applied, there isn´t any reference, any author, wy the scale at 9 points and why those descritors and not others!?. There is many ISO and works, legislation, where you can get sensory details to justify or adopt a sensorial procedure with scientific accuracy.

Some figures must be improved with better colours and line more straight, the presented are the standard of the office. The grey color adopted, also reduce the contrast in some figures, I think the the black colour is better.

In the conclusion musts be included the efecct of temperature, time, open and closed, and the physico-chemical changes, almost of the conclusion is about sensory analysis, the other results are also importants, and could influence the food safety of the chicken breast to the consumers.

Change: Line 171.… v/v %...

Author Response

Dear Reviewer,

Thank you very much for your comments and proposals, they helped a lot to improve the scientific quality of our article.

The missing p-values were added.

In order to present the measured concentrations of the individual HCAs, four tables were still incorporated.

The abstract was updated according to your proposal.

Point 2.5 was also updated by better explaining the method of sensory analysis.

The order of the keywords was changed in line with your proposal.

The conclusions were supplemented with the observed effect of temperature, time, open and closed, and the physico-chemical changes, In the part of discussion, the food safety significance of the measured HCA concentrations was included.

We also changed the v/v% in line 171.

I hope that our revised manuscript will be accepted for publication in your esteemed journal.

Yours faithfully,

Dániel Pleva

Reviewer 2 Report

The authors present manuscript with study for correlation between formation of hetero cyclic amines in the grille chicken meat temperature and time of grilling, and colored of grilled meat The HCA content was determined with validated LC-MS/MS method. The changes of meat color was measured instrumentally and visually (subjective analysis). The experiment was well designed and performed. I have no comments to the methodology part of the study. The results shows positive correlation between HCA content during the grilling and changing the color of meat. The discussion and conclusions are correct. Overall opinion about the manuscript is positive, but I miss one important thing in the paper: toxicological assessment of the HCA levels (safe, neglectful or risky?) and potential recommendation for consumers.

The photos og the grilled meat in the experiment (e.g. in the supplementary materials) can be additional value of the study I have some additional comments to the manuscript:

[Line 14 ] do not add model of the chronometer in the abstract.

[line 16] L* and a* is not understandable in the abstract-> change for "instrumental parameters" or other general definition.

[line 116 -136] The description of extract clean-up with two SPE columns in the sample prep in not clear. You divided the supernatant and clean it with two SPE columns? Please write it in simple way or add figure with sample prep scheme. [line 182] LOQ should be expressed in ng/g values.

[line 228] Figure 4 ->please add ng/g to description of y-axis.

Author Response

Dear Reviewer,

Thank you very much for your well-founded comments that were helpful to improve our manuscript.

Thank you for pointing out to the absence of the toxicological assessment of the HCA levels, that we supplemented, and also added an example photo of the meat samples we used.

We cleared the mentioned points of the abstract about L* and a*, and the sample preparation for the HPLC-MS method. Figure 4 was updated in line with your proposal, and we also corrected the unit of LOQ.

I hope that our revised manuscript will be accepted for publication in your esteemed journal.

Yours faithfully,

Dániel Pleva